# Convolutional Neural Networks on Graphs with Chebyshev Approximation, Revisited

**Mingguo He**
Renmin University of China
`mingguo@ruc.edu.cn`

**Zhewei Wei**[*]
Renmin University of China
`zhewei@ruc.edu.cn`

**Ji-Rong Wen**
Renmin University of China
`jrwen@ruc.edu.cn`

## Abstract

Designing spectral convolutional networks is a challenging problem in graph learning. ChebNet, one of the early attempts, approximates the spectral graph convolutions using Chebyshev polynomials. GCN simplifies ChebNet by utilizing only the first two Chebyshev polynomials while still outperforming it on real-world datasets. GPR-GNN and BernNet demonstrate that the Monomial and Bernstein bases also outperform the Chebyshev basis in terms of learning the spectral graph convolutions. Such conclusions are counter-intuitive in the field of approximation theory, where it is established that the Chebyshev polynomial achieves the optimum convergent rate for approximating a function.

In this paper, we revisit the problem of approximating the spectral graph convolutions with Chebyshev polynomials. We show that ChebNet's inferior performance is primarily due to illegal coefficients learnt by ChebNet approximating **analytic** filter functions, which leads to over-fitting. We then propose ChebNetII, a new GNN model based on **Chebyshev interpolation**, which enhances the original Chebyshev polynomial approximation while reducing the Runge phenomenon. We conducted an extensive experimental study to demonstrate that ChebNetII can learn arbitrary graph convolutions and achieve superior performance in both full- and semi-supervised node classification tasks. Most notably, we scale ChebNetII to a billion graph ogbn-papers100M, showing that spectral-based GNNs have superior performance. Our code is available at `https://github.com/ivam-he/ChebNetII`.

## 1 Introduction

Graph neural networks (GNNs) have received considerable attention in recent years due to their remarkable performance on a variety of graph learning tasks, including social analysis [31, 24, 38], drug discovery [49, 19, 32], traffic forecasting [26, 3, 7] and recommendation system [42, 46].

Spatial-based and spectral-based graph neural networks (GNNs) are the two primary categories of GNNs. To learn node representations, spatial-based GNNs [21, 15, 39] often rely on a message propagation and aggregation mechanism between neighboring nodes. Spectral-based methods [8, 12] create spectral graph convolutions or, equivalently, spectral graph filters, in the spectral domain of the graph Laplacian matrix. We can further divide spectral-based GNNs into two categories based on whether or not their graph convolutions can be learned.

- **Predetermined graph convolutions:** GCN [21] employs a simplified first tow Chebyshev polynomials as the graph convolution, which is a fixed low-pass filter [1, 41, 43, 54]. APPNP [22] and GDC [12] set the graph convolution with Personalized PageRank (PPR) and also achieve a low-pass filter. [12, 54]. $S^2$GC [52] derives the graph convolution from the Markov Diffusion Kernel, which is a low- and high-pass filter trade-off. GNN-LF/HF [54]

---

[*]Zhewei Wei is the corresponding author.

36th Conference on Neural Information Processing Systems (NeurIPS 2022).

designs the graph convolutions from the perspective of graph optimization that can imitate low- and high-pass filters.

- **Learnable graph convolutions:** ChebNet [8] approximates the graph convolutions using Chebyshev polynomials and, in theory, could learn arbitrary filters [1]. CayleyNet [23] learns the graph convolutions with Cayley polynomials and generates various graph filters. GPR-GNN [6] uses the Monomial basis to approximate graph convolutions, which can derive low- or high-pass filters. ARMA [2] learns the rational graph convolutions through the Auto-Regressive Moving Average filters family [28]. BernNet [17] utilizes the Bernstein basis to approximate the graph convolutions, which can also learn arbitrary graph filters.

Despite the recent developments, two fundamental problems with spectral-based GNNs remain unsolved. First of all, it is well-known that GCN [21] outperforms ChebNet [8] on real-world datasets (e.g., semi-supervised node classification tasks on citation datasets [21]). However, it is also established that GCN is a simplified version of ChebNet with only the first two Chebyshev polynomials and that ChebNet has more expressive capability than GCN in theory [1]. Consequently, a natural question is: *Why is ChebNet's performance inferior to GCN's despite its better expressiveness?*

Secondly, as shown in [17], the real-world performance of ChebNet is also inferior to that of GPR-GNN [6] and BernNet [17], which use Monomial polynomial basis and Bernstein polynomial basis to approximate the spectral graph convolutions. Such a conclusion is counter-intuitive in the field of approximation theory, where it is established that the Chebyshev polynomial achieves near-optimum error when approximating a function [13]. Therefore, the second question is: *Why is ChebNet's filter inferior to that of GPR-GNN and BernNet, despite the fact that Chebyshev polynomials have a higher approximation ability?*

In this paper, we attempt to tackle these problems by revisiting the fundamental problem of approximating the spectral graph convolutions with Chebyshev polynomials. First of all, according to the theory of the Chebyshev approximation, we observe that the coefficients of the Chebyshev expansion for an **analytic function** need to satisfy an inevitable convergence. Consequently, we prove that ChebNet's inferior performance is primarily due to illegal coefficients learnt by ChebNet approximating analytic filter functions, which leads to over-fitting. Furthermore, we propose ChebNetII, a new GNN model based on **Chebyshev interpolation**, which enhances the original Chebyshev polynomial approximation while reducing the Runge phenomenon [10]. Our ChebNetII model has robust scalability and can easily cope with various constraints on the learned filters via simple reparameterization, such as the non-negativity constraints proposed in [17]. Finally, we conduct an extensive experimental study to demonstrate that ChebNetII can achieve superior performance in both full- and semi-supervised node classification tasks and scale to the billion graph ogbn-papers100M .

## 2 Revisiting ChebNet

**Notations**. We consider an undirected graph $G = (V, E)$ with node set $V$ and edge set $E$. Let $n = |V|$ denote the number of nodes. We use $\mathbf{x} \in \mathbb{R}^n$ to denote the graph signal, where $\mathbf{x}(i)$ denotes the signal at node $i$. Note that in the general case of GNNs where the input feature is a matrix $\mathbf{X}$, we can treat each column of $\mathbf{X}$ as a graph signal. Let $\mathbf{A}$ denote the adjacency matrix and $\mathbf{D}$ denote the diagonal degree matrix, where $\mathbf{D}_{ii} = \sum_j^n \mathbf{A}_{ij}$. For convenience, we use $\mathbf{P} = \mathbf{D}^{-1/2}\mathbf{A}\mathbf{D}^{-1/2}$ and $\mathbf{L} = \mathbf{I} - \mathbf{D}^{-1/2}\mathbf{A}\mathbf{D}^{-1/2}$ to denote the normalized adjacency matrix and the normalized Laplacian matrix of $G$, respectively. We use $\mathbf{L} = \mathbf{U}\mathbf{\Lambda}\mathbf{U}^T$ to represent the eigendecomposition of $\mathbf{L}$, where $\mathbf{U}$ denotes the matrix of eigenvectors and $\mathbf{\Lambda} = diag[\lambda_1, ..., \lambda_n]$ is the diagonal matrix of eigenvalues.

### 2.1 Spectral-based GNNs and ChebNet

Spectral-based GNNs create the spectral graph convolutions in the domain of Laplacian spectrum. Recent studies suggest that many popular methods use the polynomial spectral filters to achieve graph convolutions [8, 21, 17]. We can formulate this polynomial filtering operation as

$$\mathbf{y} = \mathbf{U}diag\left[h(\lambda_1), ..., h(\lambda_n)\right]\mathbf{U}^T\mathbf{x} = \mathbf{U}h\left(\mathbf{\Lambda}\right)\mathbf{U}^T\mathbf{x} \approx \sum_{k=0}^{K} w_k\mathbf{L}^k\mathbf{x}, \tag{1}$$

where $\mathbf{y}$ denotes the filtering results of $\mathbf{x}$, and $h(\lambda)$ is called the spectral filter, which is a function of eigenvalues of the Laplacian matrix $\mathbf{L}$. The $w_k$ denote the polynomial filter weights, and the

Table 1: Comparison of ChebNet and GCN.

| Method | Cora | Citeseer | Pubmed |
|---|---|---|---|
| ChebNet (2) | $80.54_{\pm0.38}$ | $70.35_{\pm0.33}$ | $75.52_{\pm0.75}$ |
| ChebNet (10) | $74.91_{\pm0.52}$ | $67.69_{\pm0.64}$ | $65.91_{\pm1.71}$ |
| GCN | $81.32_{\pm0.18}$ | $71.77_{\pm0.21}$ | $79.15_{\pm0.18}$ |

Table 2: Comparison of different bases.

| Method | Cora | Citeseer | Pubmed |
|---|---|---|---|
| ChebBase | $79.29_{\pm0.36}$ | $70.76_{\pm0.37}$ | $78.07_{\pm0.32}$ |
| GPR-GNN | $83.95_{\pm0.22}$ | $70.92_{\pm0.57}$ | $78.97_{\pm0.27}$ |
| BernNet | $83.15_{\pm0.32}$ | $72.24_{\pm0.25}$ | $79.65_{\pm0.25}$ |

polynomial filter can be defined as $h(\lambda) = \sum_{k=0}^{K} w_k \lambda^k, \lambda \in [0, 2]$. ChebNet [8] is a remarkable attempt in this field, which uses Chebyshev polynomial to approximate the filtering operation.

$$\mathbf{y} \approx \sum_{k=0}^{K} w_k T_k(\hat{\mathbf{L}})\mathbf{x}, \tag{2}$$

where $\hat{\mathbf{L}} = 2\mathbf{L}/\lambda_{max} - \mathbf{I}$ denotes the scaled Laplacian matrix. $\lambda_{max}$ is the largest eigenvalue of $\mathbf{L}$ and $w_k$ denote the Chebyshev coefficients. The Chebyshev polynomials can be recursively defined as $T_k(x) = 2x T_{k-1}(x) - T_{k-2}(x)$, with $T_0(x) = 1$ and $T_1(x) = x$. ChebNet's structure is:

$$\mathbf{Y} = \sum_{k=0}^{K} T_k(\hat{\mathbf{L}})\mathbf{X}\mathbf{W}_k, \tag{3}$$

with the trainable weights $\mathbf{W}_k$. The Chebyshev coefficients $w_k$ of the filtering operation (2) are implicitly encoded in the weight matrices $\mathbf{W}_k$. We list more spectral-based GNNs' details in the supplementary materials.

## 2.2 The motivation of revisiting ChebNet

**ChebNet versus GCN.** Even though GCN is a simplified form of ChebNet, it is well known that ChebNet is inferior to GCN for semi-supervised node classification tasks [21]. Table 1 shows the results of ChebNet and GCN for semi-supervised node classification tasks on Cora, Citeseer and Pubmed datasets (see the Appendix for experimental details) . We find that ChebNet is inferior to GCN, especially when we increase the polynomial order $K$ from 2 to 10 in Equation (3).

On the other hand, existing research [1] has shown that ChebNet is more expressive than GCN in theory. In particular, ChebNet can approximate arbitrary spectral filters as $K$ increases, while GCN is a fixed low-pass filter. If we set $K = 1$ and $w_0 = w_1$ in the Equation (2), ChebNet corresponds to a high-pass filter; if we set $K = 1$ and $w_0 = -w_1$, ChebNet becomes a low-pass filter which is essentially the same as GCN. Consequently, a natural question is: *Why is ChebNet's performance inferior to GCN's despite its better expressiveness?*

**Chebyshev basis versus other bases.** Chebyshev polynomials are widely used to approximate various functions in the digital signal processing and the graph signal filtering [36, 37]. The truncated Chebyshev expansions are demonstrated to produce a minimax polynomial approximation for the analytic functions [13]. Consequently, the spectral filters can be well-approximated by a truncated expansion in terms of Chebyshev polynomials $T_k(x)$ up to $K$-th order [16].

$$h(\hat{\lambda}) \approx \sum_{k=0}^{K} w_k T_k(\hat{\lambda}), \hat{\lambda} \in [-1, 1], \tag{4}$$

where $\hat{\lambda}$ is the eigenvalue of the scaled Laplacian matrix $\hat{\mathbf{L}}$. ChebNet [8] then defined the graph convolutions using the Chebyshev approximated filters, while recent works were inspired by ChebNet and used Monomial (i.e., GPR-GNN [6]) and Bernstein (i.e., BernNet [17]) bases to approximate filters. In order to evaluate the approximation ability of Chebyshev basis, we propose ChebNet with explicit coefficients, ChebBase, which simply replaces the Monomial basis of GPR-GNN and Bernstein basis of BernNet with the Chebyshev basis. The expression of ChebBase is

$$\mathbf{Y} = \sum_{k=0}^{K} w_k T_k(\hat{\mathbf{L}})f_\theta(\mathbf{X}), \tag{5}$$

where $f_\theta(\mathbf{X})$ denotes Multi-Layer Perceptron (MLP). Table 2 reveals the results of ChebBase, GPR-GNN and BernNet for node classification tasks on three citation graphs. We can observe that

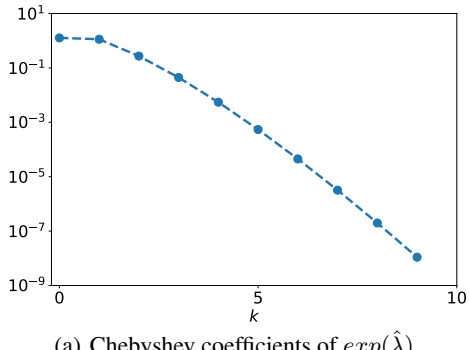
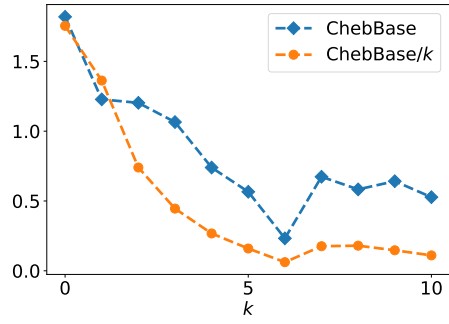

(a) Chebyshev coefficients of $exp(\hat{\lambda})$.  (b) Chebyshev coefficients learnt on Cora.

Figure 1: Illustrations of the Chebyshev expansion's coefficients of $exp(\hat{\lambda})$ and the Chebyshev coefficients learnt by ChebBase and ChebBase/$k$ on Cora.

ChebBase has the worst performance, which is inconsistent with the fact that the Chebyshev basis can approximate minimax polynomial in theory. Therefore, the second question is: *Why is ChebNet's filter inferior to that of GPR-GNN and BernNet, despite the fact that Chebyshev polynomials have a higher approximation ability?*

### 2.3 Coefficient Constraints

We now demonstrate that ChebNet's suboptimal performance is due to the illegal coefficients learned, which results in over-fitting. Given an arbitrary continuous function $f(x)$ in the interval $[-1, 1]$, the Chebyshev expansion is defined as $f(x) = \sum_{k=0}^{\infty} w_k T_k(x)$ with the Chebyshev coefficients $w_k$. The following theorem establishes that in order to approximate an analytic function, the Chebyshev expansion's coefficients must be constrained.

**Theorem 2.1.** [48] *If $f(x)$ is weakly singular at the boundaries and analytic in the interval $(-1, 1)$, then the Chebyshev coefficients $w_k$ will asymptotically (as $k \to \infty$) decrease proportionally to $1/k^q$ for some positive constant q.*

Here, "weakly singular" means that the derivative of $f$ could vanish at the boundaries, and "analytic" means $f$ can be locally given by a convergent power series in the interval $(-1, 1)$. Intuitively, Chebyshev polynomial $T_k(x)$ with larger $k$ corresponds to higher frequency oscillation (see the Appendix for more details). Theorem 2.1 essentially demonstrates that high frequency polynomials should be constrained in the Chebyshev expansion to approximate an analytic function. Figure 1(a) depicts the Chebyshev expansion's coefficients of the analytic function $exp(\hat{\lambda})$ used as a spectral filter in GDC [12] and shows that the coefficients are convergent.

The ability to approximate an analytic function is crucial in the task of approximating the spectral filters, since non-analytic filters are more difficult to approximate by polynomials and may result in over-fitting. In particular, ChebNet and ChebBase learn the coefficients $w_k$ by gradient descent without any constraints. The coefficients may not satisfy Theorem 2.1, leading to their poor performance. To

Table 3: The performance of ChebBase.

| Method | Cora | Citeseer | Pubmed |
|---|---|---|---|
| ChebNet | $80.54_{\pm 0.38}$ | $70.35_{\pm 0.33}$ | $75.52_{\pm 0.75}$ |
| GCN | $81.32_{\pm 0.18}$ | $71.77_{\pm 0.21}$ | $79.15_{\pm 0.18}$ |
| ChebBase | $79.29_{\pm 0.36}$ | $70.76_{\pm 0.37}$ | $78.07_{\pm 0.32}$ |
| ChebBase/$k$ | $\mathbf{82.66_{\pm 0.28}}$ | $\mathbf{72.52_{\pm 0.29}}$ | $\mathbf{79.25_{\pm 0.31}}$ |

validate this conjuncture, we conducted an empirical analysis of ChebBase with difference coefficient constraints. Inspired by Theorem 2.1, we use the following propagation process for the ChebBase/$k$.

$$\mathbf{Y} = \sum_{k=0}^{K} \frac{w_k}{k} T_k(\hat{\mathbf{L}}) f_\theta(\mathbf{X}), \tag{6}$$

where $w_k/k$ denote the Chebyshev coefficients implemented by reparameterizing the learnable parameters $w_k$. Table 3 shows the experimental results of the semi-supervised node classification performed on the citation graphs. We can observe that with a simple penalty on $w_k$, ChebBase/$k$ outperforms ChebNet, ChebBase, and GCN. Figure 1(b) plots the absolute value of the Chebyshev coefficients learnt by ChebBase and ChebBase/$k$ on Cora. We can observe that the coefficients of ChebBase/$k$ could more readily satisfy the convergence constraint. These results validate Theorem 2.1.

# 3 ChebNetII model

Although ChebBase/$k$ appears to be a promising approach, it still has some drawbacks: 1) Imposing the penalty on the coefficients is not mathematically elegant, as Theorem 2.1 only provides a necessary condition for the coefficients; 2) It is hard to impose further constraints on the learned spectral filters. For example, it is unclear how we can modify Equation (6) to obtain non-negative filters, a requirement proposed in [17]. In this section, we describe ChebNetII, a GNN model based on Chebyshev interpolation that resolves the above two issues. We also discuss the advantages and disadvantages of various polynomial interpolations as well as the Runge phenomenon.

## 3.1 Chebyshev interpolation

Consider a real filter function $h(\hat{\lambda})$ that is continuous in the interval $[-1, 1]$. When the values of this filter are known at a finite number of points $\hat{\lambda}_k$, one can consider the approximation by a polynomial $P_K$ with $K$ degree such that $h(\hat{\lambda}_k) = P_K(\hat{\lambda}_k)$, which is the general polynomial interpolation. We give an explicit expression of the general polynomial interpolation in the supplementary materials.

We generally sample the $K + 1$ points $\hat{\lambda}_0 < \hat{\lambda}_1 < ... < \hat{\lambda}_K$ uniformly from $[-1, 1]$ to construct the interpolating polynomial $P_K(\hat{\lambda})$. Intuitively, increasing $K$ should improve the approximation quality. However, this is not always the case due to the Runge Phenomenon [10] (The details are discussed in section 3.3). The popular approach to this problem in the literature [14] is Chebyshev interpolation, having superior approximation ability and faster convergence. Instead of sampling the interpolation points uniformly, Chebyshev interpolation uses Chebyshev nodes as the interpolation points, which are essentially the zeros of the $(K + 1)$-th Chebyshev polynomial.

**Definition 3.1.** (**Chebyshev Nodes**) *The Chebyshev polynomial $T_k(x)$ satisfies the closed form expression $T_k(x) = \cos(k \arccos(x))$. The Chebyshev Nodes for $T_k(x)$ are defined as $x_j = \cos\left(\frac{2j+1}{2k}\pi\right), j = 0, 1, ..., k - 1$, which lie in the interval $(-1, 1)$ and are the zeros of $T_k(x)$.*

Definition 3.1 suggests that each Chebyshev polynomial $T_k(x)$ has $k$ zeros, and we can define Chebyshev interpolation by replacing the equispaced points with Chebyshev nodes in the general polynomial interpolation (see the Appendix for details). More eloquently, definition 3.2 efficiently defines the Chebyshev interpolation via their orthogonality properties.

**Definition 3.2.** (**Chebyshev Interpolation**) [14] *Given a continuous filter function $h(\hat{\lambda})$, let $x_j = cos\left(\frac{j+1/2}{K+1}\pi\right), j = 0, \ldots, K$ denote the Chebyshev nodes for $T_{K+1}$ and $h(x_j)$ denotes the function value at node $x_j$. The Chebyshev interpolation of $h(\hat{\lambda})$ is defined to be*

$$P_K(\hat{\lambda}) = \sum_{k=0}^{K} c'_k T_k(\hat{\lambda}), c_k = \frac{2}{K+1} \sum_{j=0}^{K} h(x_j) T_k(x_j), \tag{7}$$

*where the prime indicates the first term is to be halved, that is, $c'_0 = c_0/2$, $c'_1 = c_1, \ldots, c'_K = c_K$.*

## 3.2 ChebNetII via Chebyshev Interpolation

Inspired by Chebyshev interpolation, we propose ChebNetII, a graph convolutional network that approximates an arbitrary spectral filter $h(\hat{\lambda})$ with an optimal convergence rate. ChebNetII simply reparameterizes the filter value $h(x_j)$ in Equation (7) as a learnable parameter $\gamma_j$, which allows the model to learn an arbitrary spectral filter via gradient descent. More precisely, the **ChebNetII** model can be formulated as

$$\mathbf{Y} = \frac{2}{K+1} \sum_{k=0}^{K} \sum_{j=0}^{K} \gamma_j T_k(x_j) T_k(\hat{\mathbf{L}}) f_\theta(\mathbf{X}), \tag{8}$$

where $x_j = cos\left((j + 1/2)\pi/(K+1)\right)$ are the Chebyshev nodes of $T_{K+1}$, $f_\theta(\mathbf{X})$ denotes an MLP on the node feature matrix $\mathbf{X}$, and $\gamma_j$ for $j = 0, 1, ..., K$ are the learnable parameters. Note that similar to APPNP [22], we decouple feature propagation and transformation.

Table 4: Dataset statistics.

| | Chameleon | Squirrel | Actor | Texas | Cornell | Cora | Citeseer | Pubmed | ogbn-arxiv | ogbn-papers100M |
|---|---|---|---|---|---|---|---|---|---|---|
| Nodes | 2277 | 5201 | 7600 | 183 | 183 | 2708 | 3327 | 19,717 | 169,343 | 111,059,956 |
| Edges | 31,371 | 198,353 | 26,659 | 279 | 277 | 5278 | 4552 | 44,324 | 1,166,243 | 1,615,685,872 |
| Features | 2325 | 2089 | 932 | 1703 | 1703 | 1433 | 3703 | 500 | 128 | 128 |
| Classes | 5 | 5 | 5 | 5 | 5 | 7 | 6 | 5 | 40 | 172 |
| $\mathcal{H}(G)$ | 0.23 | 0.22 | 0.22 | 0.11 | 0.30 | 0.81 | 0.74 | 0.80 | 0.66 | - |

Consequently, the filtering operation of ChebNetII can be expressed as

$$\mathbf{y} \approx \frac{2}{K+1} \sum_{k=0}^{K} \sum_{j=0}^{K} \gamma_j T_k(x_j) T_k(\hat{\mathbf{L}}) \mathbf{x}. \tag{9}$$

It is easy to see that compared to the filtering operation of the original ChebNet (2), we only make one simple change: reparameterizing the coefficient $w_k$ by $w_k = \frac{2}{K+1} \sum_{j=0}^{K} \gamma_j T_k(x_j)$. However, this simple modification allows us to have more control on shaping the resulting filter, as Chebyshev interpolation suggests that $\gamma_j$ directly corresponds to the filter value $h(x_j)$ at the Chebyshev node $x_j$. The coefficients $w_k = \frac{2}{K+1} \sum_{j=0}^{K} h(x_j) T_k(x_j)$ are fundamentally guaranteed to satisfy the constraints of Theorem 2.1 since we directly approximate the filter $h$. Furthermore, Chebyshev interpolation also provides the ChebNetII with several beneficial mathematical properties.

### 3.3 Analysis of ChebNetII

ChebNetII has several advantages over existing GNN models due to the unique nature of Chebyshev interpolation. From the standpoint of polynomial approximation and computational complexity, we compare ChebNetII with current related approaches such as GPR-GNN [6] and BernNet [17].

**Near-minimax approximation.** First of all, we examine ChebNetII's capabilities in terms of filter function approximation. Theorem 3.1 exhibits that ChebNetII provides an approximation that is close to the best polynomial approximation for a spectral filter $h$.

**Theorem 3.1.** [27] *A polynomial approximation $P_K^*(x)$ for a function $f(x)$ is said to be near-best/minimax approximation with a relative distance $\rho$ if*

$$||f(x) - P_K^*(x)|| \le (1 + \rho)||f(x) - P_B^*(x)||, \tag{10}$$

*where $\rho$ is the Lebesgue constant, $P_B^*(x)$ is a best polynomial approximation, and $|| \cdot ||$ represents the uniform norm (i.e., $||g|| = \max_{x \in [-1,1]} |g(x)|$). Then, we have $\rho \sim 2^K$ as $K \to \infty$ for the general polynomial interpolation, and $\rho \sim \log(K)$ as $K \to \infty$ for the Chebyshev interpolation.*

**Convergence.** In comparison to BernNet [17], which uses the Bernstein basis, ChebNetII has a faster convergence rate for approximating a filter function. Specifically, we have the following Theorem:

**Theorem 3.2.** [14, 29] *Let $P_K(x)$ be the polynomial approximation for a function $f(x)$. Then the error is given as $||f(x) - P_K(x)|| \le E(K)$. If $P_K(x)$ is obtained by Bernstein approximation, then $E(K) \sim (1 + (2K)^{-2})\omega(K^{-1/2})$; if $P_K(x)$ is obtained by Chebyshev Interpolation, then $E(K) \sim C\omega(K^{-1})\log(K)$ with a constant $C$, where $\omega$ is the modulus of continuity.*

**Runge phenomenon.** In comparison to GPR-GNN [6], which uses the Monomial basis, ChebNetII has the advantage of reducing the Runge phenomenon [10]. In particular, when we use the general polynomial interpolation to approximate a Runge filter $h(\hat{\lambda})$ with a high degree over a set of equis-paced interpolation points, it will cause oscillation along the edges of an interval. Consequently, as the degree of the polynomial increases, the interpolation error increases. Following [14], we define the error of polynomial interpolation as

$$R_K(\hat{\lambda}) = h(\hat{\lambda}) - P_K(\hat{\lambda}) = \frac{h^{K+1}(\zeta)}{(K+1)!} \pi_{K+1}(\hat{\lambda}), \tag{11}$$

where $\pi_{K+1}(\hat{\lambda}) = \prod_{k=0}^{K}(\hat{\lambda} - \hat{\lambda}_k)$ denotes the nodal polynomial and $\zeta$ is the value depending on $\hat{\lambda}$. The terrible Runge phenomenon is caused by the values of this nodal polynomial, which have very

Table 5: Dataset statistics of large heterophilic graphs.

| | Penn94 | pokec | arXiv-year | genius | twitch-gamers | wiki |
|---|---|---|---|---|---|---|
| Nodes | 41,554 | 1,632,803 | 169,343 | 421,961 | 168,114 | 1,925,342 |
| Edges | 1,362,229 | 30,622,564 | 1,166,243 | 984,979 | 6,797,557 | 303,434,860 |
| Features | 5 | 65 | 128 | 12 | 7 | 600 |
| Classes | 2 | 2 | 5 | 2 | 2 | 5 |
| $\mathcal{H}(G)$ | 0.47 | 0.46 | 0.22 | 0.62 | 0.55 | 0.39 |

high oscillations around the interval endpoints. In particular, for high-degree polynomial interpolation at equidistant points in $[-1, 1]$, we have $\lim_{K \to \infty} \left( \max_{-1 \leq \hat{\lambda} \leq 1} |R_K(\hat{\lambda})| \right) = \infty$.

On the other hand, we have the following Theorem 3.3 that explains that Chebyshev nodes can minimize and quantify this error caused by the nodal polynomial, meaning Chebyshev interpolation minimizes the problem of the Runge phenomenon.

**Theorem 3.3.** [14] *Consider the Chebyshev nodes* $x_j = \cos((j + 1/2)\pi/(K + 1))$, $j = 0, 1, ..., K$. *Then the nodal polynomial* $\hat{T}_{K+1}(x) = \prod_{k=0}^{K}(x - x_j)$ *has the smallest possible uniform norm, i.e.,* $||\hat{T}_{K+1}(x)|| = 2^{-K}$.

**Computational complexity.** Compared to BernNet [17], which has a time complexity quadratic to the order $K$ in the forward process, ChebNetII can be computed in time linear to $K$. Specifically, we first compute the ChebNetII's coefficients $\frac{2}{K+1} \sum_{j=0}^{K} \gamma_j T_k(x_j)$ in time linear to $K$ as we can precompute $T_k(x_j)$, and then plug the coefficients into Equation (8) for propagation, which also takes the time linear to $K$, the same as that of ChebNet [8] and GPR-GNN [6].

## 4 Experiments

In this section, we conduct experiments to evaluate the performance of ChebNetII against the state-of-the-art graph neural networks on a wide variety of open graph datasets.

**Dataset and Experimental setup.** We evaluate ChebNetII on several real-world graphs for the Semi- and Full-supervised node classification tasks. The datasets include three homophilic citation graphs: Cora, Citeseer, and Pubmed [35, 45], five heterophilic graphs: the Wikipedia graphs Chameleon and Squirrel [34], the Actor co-occurrence graph, and webpage graphs Texas and Cornell from WebKB* [30], two large citation graphs: ogbn-arxiv and ogbn-papers100M [18], as well as six large heterophilic graphs: Penn94, pokec, arXiv-year, genius, twitch-gamers and wiki [25]. We measure the level of homophily in a graph using the edge homophily ratio $\mathcal{H}(G) = \frac{|\{(u,v):(u,v) \in E \wedge y_v = y_u\}|}{|E|}$ [53], where $y_v$ denotes the label of node $v$. We summarize the dataset statistics in Tables 4 and 5. All the experiments are carried out on a machine with an NVIDIA RTX8000 GPU (48GB memory), Intel Xeon CPU (2.20 GHz) with 40 cores, and 512 GB of RAM.

### 4.1 Semi-supervised node classification with polynomial-based methods

**Setting and baselines.** For the semi-supervised node classification task, we compare ChebNetII to 7 polynomial approximation filter methods, including MLP, GCN [21], ARMA [2], APPNP [22], ChebNet [8], GPR-GNN [6] and BernNet [17]. For dataset splitting, we employ both random and fixed splits and report the results on random splits. The results of fixed splits will be discussed in the Appendix.Specifically, we apply the standard training/validation/testing split [45] on the three homophilic citation datasets (i.e., Cora, Citeseer, and Pubmed), with 20 nodes per class for training, 500 nodes for validation, and 1,000 nodes for testing. Since this standard split can not be used for very small graphs (e.g. Texas), we use the sparse splitting [6] with the training/validation/test sets accounting for 2.5%/2.5%/95%, respectively, on the five heterophilic datasets.

For ChebNetII, we use Equation (8) as the propagation process and use the $ReLu$ function to reparametrize $\gamma_j$, maintaining the non-negativity of the filters [17]. We set the hidden units as 64

---

*http://www.cs.cmu.edu/afs/cs.cmu.edu/project/theo-11/www/wwkb/

Table 6: Mean classification accuracy of **semi-supervised** node classification with random splits.

| Method | Cham. | Squi. | Texas | Corn. | Actor | Cora | Cite. | Pubm. |
|---|---|---|---|---|---|---|---|---|
| MLP | $26.36_{\pm2.85}$ | $21.42_{\pm1.50}$ | $32.42_{\pm9.91}$ | $36.53_{\pm7.92}$ | $29.75_{\pm0.95}$ | $57.17_{\pm1.34}$ | $56.75_{\pm1.55}$ | $70.52_{\pm2.01}$ |
| GCN | $38.15_{\pm3.77}$ | $31.18_{\pm0.93}$ | $34.68_{\pm9.07}$ | $32.36_{\pm8.55}$ | $22.74_{\pm2.37}$ | $79.19_{\pm1.37}$ | $69.71_{\pm1.32}$ | $78.81_{\pm0.84}$ |
| ChebNet | $37.15_{\pm1.49}$ | $26.55_{\pm0.46}$ | $36.35_{\pm8.90}$ | $28.78_{\pm4.85}$ | $26.58_{\pm1.92}$ | $78.08_{\pm0.86}$ | $67.87_{\pm1.49}$ | $73.96_{\pm1.68}$ |
| ARMA | $37.42_{\pm1.72}$ | $24.15_{\pm0.93}$ | $39.65_{\pm8.09}$ | $28.90_{\pm10.07}$ | $27.02_{\pm2.31}$ | $79.14_{\pm1.07}$ | $69.35_{\pm1.44}$ | $78.31_{\pm1.33}$ |
| APPNP | $32.73_{\pm2.31}$ | $24.50_{\pm0.89}$ | $34.79_{\pm10.11}$ | $34.85_{\pm9.71}$ | $29.74_{\pm1.04}$ | $82.39_{\pm0.68}$ | $69.79_{\pm0.92}$ | $\mathbf{79.97_{\pm1.58}}$ |
| GPR-GNN | $33.03_{\pm1.92}$ | $24.36_{\pm1.52}$ | $33.98_{\pm11.90}$ | $38.95_{\pm12.36}$ | $28.58_{\pm1.01}$ | $82.37_{\pm0.91}$ | $69.22_{\pm1.27}$ | $79.28_{\pm2.25}$ |
| BernNet | $27.32_{\pm4.04}$ | $22.37_{\pm0.98}$ | $43.01_{\pm7.45}$ | $39.42_{\pm9.59}$ | $29.87_{\pm0.78}$ | $82.17_{\pm0.86}$ | $69.44_{\pm0.97}$ | $79.48_{\pm1.47}$ |
| ChebNetII | $\mathbf{43.42_{\pm3.54}}$ | $\mathbf{33.96_{\pm1.22}}$ | $\mathbf{46.58_{\pm7.68}}$ | $\underline{42.19_{\pm11.61}}$ | $\mathbf{30.18_{\pm0.81}}$ | $\mathbf{82.42_{\pm0.64}}$ | $\mathbf{69.89_{\pm1.21}}$ | $\underline{79.51_{\pm1.03}}$ |

Table 7: Mean classification accuracy of **full-supervised** node classification with random splits.

| Method | Cham. | Squi. | Texas | Corn. | Actor | Cora | Cite. | Pubm. |
|---|---|---|---|---|---|---|---|---|
| MLP | $46.59_{\pm1.84}$ | $31.01_{\pm1.18}$ | $86.81_{\pm2.24}$ | $84.15_{\pm3.05}$ | $40.18_{\pm0.55}$ | $76.89_{\pm0.97}$ | $76.52_{\pm0.89}$ | $86.14_{\pm0.25}$ |
| GCN | $60.81_{\pm2.95}$ | $45.87_{\pm0.88}$ | $76.97_{\pm3.97}$ | $65.78_{\pm4.16}$ | $33.26_{\pm1.15}$ | $87.18_{\pm1.12}$ | $79.85_{\pm0.78}$ | $86.79_{\pm0.31}$ |
| ChebNet | $59.51_{\pm1.25}$ | $40.81_{\pm0.42}$ | $86.28_{\pm2.62}$ | $83.91_{\pm2.17}$ | $37.42_{\pm0.58}$ | $87.32_{\pm0.92}$ | $79.33_{\pm0.57}$ | $87.82_{\pm0.24}$ |
| ARMA | $60.21_{\pm1.00}$ | $36.27_{\pm0.62}$ | $83.97_{\pm3.77}$ | $85.62_{\pm2.13}$ | $37.67_{\pm0.54}$ | $87.13_{\pm0.80}$ | $80.04_{\pm0.55}$ | $86.93_{\pm0.24}$ |
| APPNP | $52.15_{\pm1.79}$ | $35.71_{\pm0.78}$ | $90.64_{\pm1.70}$ | $91.52_{\pm1.81}$ | $39.76_{\pm0.49}$ | $88.16_{\pm0.74}$ | $80.47_{\pm0.73}$ | $88.13_{\pm0.33}$ |
| GCNII | $63.44_{\pm0.85}$ | $41.96_{\pm1.02}$ | $80.46_{\pm5.91}$ | $84.26_{\pm2.13}$ | $36.89_{\pm0.95}$ | $88.46_{\pm0.82}$ | $79.97_{\pm0.65}$ | $\mathbf{89.94_{\pm0.31}}$ |
| TWIRLS | $50.21_{\pm2.97}$ | $39.63_{\pm1.02}$ | $91.31_{\pm3.36}$ | $89.83_{\pm2.29}$ | $38.13_{\pm0.81}$ | $88.57_{\pm0.91}$ | $80.07_{\pm0.94}$ | $88.87_{\pm0.43}$ |
| EGNN | $51.55_{\pm1.73}$ | $35.81_{\pm0.91}$ | $81.34_{\pm1.56}$ | $82.09_{\pm1.16}$ | $35.16_{\pm0.64}$ | $87.47_{\pm1.33}$ | $80.51_{\pm0.93}$ | $88.74_{\pm0.46}$ |
| PDE-GCN | $66.01_{\pm1.56}$ | $48.73_{\pm1.06}$ | $93.24_{\pm2.03}$ | $89.73_{\pm1.35}$ | $39.76_{\pm0.74}$ | $88.62_{\pm1.03}$ | $\underline{79.98_{\pm0.97}}$ | $89.92_{\pm0.38}$ |
| GPR-GNN | $67.49_{\pm1.38}$ | $50.43_{\pm1.89}$ | $\underline{92.91_{\pm1.32}}$ | $91.57_{\pm1.96}$ | $39.91_{\pm0.62}$ | $\underline{88.54_{\pm0.67}}$ | $80.13_{\pm0.84}$ | $\underline{88.46_{\pm0.31}}$ |
| BernNet | $68.53_{\pm1.68}$ | $51.39_{\pm0.92}$ | $92.62_{\pm1.37}$ | $92.13_{\pm1.64}$ | $41.71_{\pm1.12}$ | $88.51_{\pm0.92}$ | $80.08_{\pm0.75}$ | $88.51_{\pm0.39}$ |
| ChebNetII | $\mathbf{71.37_{\pm1.01}}$ | $\mathbf{57.72_{\pm0.59}}$ | $\mathbf{93.28_{\pm1.47}}$ | $\mathbf{92.30_{\pm1.48}}$ | $\mathbf{41.75_{\pm1.07}}$ | $\mathbf{88.71_{\pm0.93}}$ | $\mathbf{80.53_{\pm0.79}}$ | $88.93_{\pm0.29}$ |

and $K = 10$ for the all datasets as the same as GPR-GNN [6] and BernNet [17]. We employ the Adam SGD optimizer [20] with an early stopping of 200 and a maximum of 1000 epochs to train ChebNetII. We use the officially released code for GPR-GNN and BernNet and use the Pytorch Geometric library implementations [11] for other models (i.e., MLP, GCN, APPNP, ARMA, and ChebNet). More details of hyper-parameters and baselines' settings are listed in the Appendix.

**Results.** We utilize accuracy (the micro-F1 score) with a 95% confidence interval as the evaluation metric. Table 6 reports the relevant results on 10 random splits. Boldface letters indicate the best result for the given confidence interval, and underlinings denote the next best result. We first observe that ChebNet is inferior to GCN even on heterophilic graphs, which concurs with our theoretical analysis that the illegal coefficients learned by ChebNet lead to over-fitting. ChebNetII, on the other hand, outperforms other methods on all datasets excluding Pubmed, where it also achieves top-2 classification accuracy. This quality is due to the fact that the learnable parameters $\gamma_j$ of ChebNetII directly correspond to the filter value $h(x_j)$ at the Chebyshev node $x_j$, effectively preventing it from learning an illegal filter.

## 4.2   Full-supervised node classification

**Setting and baselines.** For full-supervised node classification, we compare ChebNetII to the baselines in the prior semi-supervised node classification. We also include GCNII [5], TWIRLS [44], EGNN [51] and PDE-GCN [9] four competitive baselines for full-supervised node classification. For all datasets, we randomly split the nodes into 60%, 20%, and 20% for training, validation and testing, and all methods share the same 10 random splits for a fair comparison, as suggested in [30, 6, 17].

For ChebNetII, we also set the hidden units to be 64 and $K = 10$ for all datasets, and employ the same training manner as in the semi-supervised node classification task. For GCNII, TWIRLS, EGNN and PDE-GCN we use the officially released code. More details of hyper-parameters and baselines' settings are listed in the Appendix.

**Results.** Table 7 reports the mean classification accuracy of each model. We first observe that, given more training data, ChebNet starts to outperform GCN on both homophilic and heterophilic datasets, which demonstrates the effectiveness of the Chebyshev approximation. However, we also observe that ChebNetII achieves new state-of-the-art results on 7 out of 8 datasets and competitive results on Pubmed. Notably, ChebNetII outperforms GPR-GNN and BernNet by over 10% on the

Table 8: Experimental results on large heterophilic graphs. OOM denotes "out of memory".

| Method | Penn94 | pokec | arXiv-year | genius | twitch-gamers | wiki |
|---|---|---|---|---|---|---|
| MLP | $73.61_{\pm 0.40}$ | $62.37_{\pm 0.02}$ | $36.70_{\pm 0.21}$ | $86.68_{\pm 0.09}$ | $60.92_{\pm 0.07}$ | $37.38_{\pm 0.21}$ |
| LINK | $80.79_{\pm 0.49}$ | $80.54_{\pm 0.03}$ | $53.97_{\pm 0.18}$ | $73.56_{\pm 0.14}$ | $64.85_{\pm 0.21}$ | $57.11_{\pm 0.26}$ |
| LINKX | $84.71_{\pm 0.52}$ | $82.04_{\pm 0.07}$ | $\overline{56.00}_{\pm 1.34}$ | $90.77_{\pm 0.27}$ | $\mathbf{66.06_{\pm 0.19}}$ | $\underline{59.80}_{\pm 0.41}$ |
| GCN | $\overline{82.47}_{\pm 0.27}$ | $\overline{75.45}_{\pm 0.17}$ | $46.02_{\pm 0.26}$ | $\overline{87.42}_{\pm 0.37}$ | $62.18_{\pm 0.26}$ | OOM |
| GCNII | $82.92_{\pm 0.59}$ | $78.94_{\pm 0.11}$ | $47.21_{\pm 0.28}$ | $90.24_{\pm 0.09}$ | $63.39_{\pm 0.61}$ | OOM |
| ChebNet | $82.59_{\pm 0.31}$ | $72.71_{\pm 0.66}$ | $46.76_{\pm 0.24}$ | $89.36_{\pm 0.31}$ | $62.31_{\pm 0.37}$ | OOM |
| GPR-GNN | $83.54_{\pm 0.32}$ | $80.74_{\pm 0.22}$ | $45.97_{\pm 0.26}$ | $90.15_{\pm 0.30}$ | $62.59_{\pm 0.38}$ | $58.73_{\pm 0.34}$ |
| BernNet | $83.26_{\pm 0.29}$ | $81.67_{\pm 0.17}$ | $46.34_{\pm 0.32}$ | $90.47_{\pm 0.33}$ | $64.27_{\pm 0.31}$ | $59.02_{\pm 0.29}$ |
| ChebNetII | $\mathbf{84.86_{\pm 0.33}}$ | $\mathbf{82.33_{\pm 0.28}}$ | $48.53_{\pm 0.31}$ | $\mathbf{90.85_{\pm 0.32}}$ | $\underline{65.03}_{\pm 0.27}$ | $\mathbf{60.95_{\pm 0.39}}$ |

Squirrel dataset. We attribute this quality to the fact that Chebyshev interpolation achieves near-minimax approximation of any function with respect to the uniform norm, giving ChebNetII greater approximation power than GPR-GNN and BernNet do.

## 4.3 Scalability of ChebNetII

For ChebNetII, if we calculate and save $T_k(\hat{\mathbf{L}})\mathbf{X}$ for $k \in 0, \cdots, K$ in the preprocessing, we can scale it to large graphs. Specifically, we use the below propagation expression.

$$\mathbf{Y} = f_\theta(\mathbf{Z}), \quad \mathbf{Z} = \frac{2}{K+1} \sum_{k=0}^{K} \sum_{j=0}^{K} \gamma_j T_k(x_j) T_k(\hat{\mathbf{L}})\mathbf{X}. \tag{12}$$

The pre-computed $\hat{\mathbf{L}}^k \mathbf{X}$ allow us to train $\gamma_j$ and $f_\theta(\cdot)$ in a mini-batch manner. This approach also works for GPR-GNN [6] and BernNet [17], so we report their results in this manner when ChebNetII does. We evaluate the scalability of ChebNetII on the large heterophilic graphs [25] and the widely used OGB datasets [18].

On the large heterophilic graphs, we compare Cheb-NetII to eight competitive baselines, including MLP, LINK [50], LINKX [25], GCN [21], GCNII [5], ChebNet [8], GPR-GNN [6] and BernNet [17]. For ChebNetII, we use Equation (12) as the propagation process on pokec and wiki and Equation (8) on the four remaining datasets. We establish the experimental setting following [25] and use the published baselines' results, excluding GPR-GNN. More details are listed in the Appendix. Table 8 reports the mean results of each method over 5 runs. ChebNetII outperforms all other methods on 4 out of 6 datasets and the polynomial-based methods on arXiv-year and twitch-gamers. Notably, LINK and LINKX outperform ChebNetII on arXiv-year because they use a directed graph on this dataset. Using the directed

Table 9: Mean classification accuracy on large graphs. OOM denotes "out of memory" and "-" means failing to finish preprocessing in 24h.

| Method | ogbn-arxiv | ogbn-papers100M |
|---|---|---|
| GCN | $71.74_{\pm 0.29}$ | OOM |
| ChebNet | $71.12_{\pm 0.22}$ | OOM |
| ARMA | $71.47_{\pm 0.25}$ | OOM |
| GPR-GNN | $71.78_{\pm 0.18}$ | $65.89_{\pm 0.35}$ |
| BernNet | $71.96_{\pm 0.27}$ | - |
| SIGN | $71.95_{\pm 0.12}$ | $65.68_{\pm 0.16}$ |
| GBP | $72.21_{\pm 0.17}$ | $65.23_{\pm 0.31}$ |
| NDLS* | $\underline{72.24}_{\pm 0.21}$ | $\underline{65.61}_{\pm 0.29}$ |
| ChebNetII | $\mathbf{72.32_{\pm 0.23}}$ | $\mathbf{67.18_{\pm 0.32}}$ |

graphs to the spectral-based GNNs is a future meaningful work because the current spectral graph theory only applies to undirected graphs. For the largest heterophilic graph wiki, ChebNetII obtains a new state-of-the-art result. We attribute that ChebNetII can precompute $\hat{\mathbf{L}}^k \mathbf{X}$ without the graph sampling used in LINK and LINKX and has a strong filter approximation ability.

On ogbn-arxiv and -papers100M, we compare ChebNetII to polynomial-based GNNs and state-of-the-art scalable GNNs, SIGN [33], GBP [4], and NDLS* [47]. We follow the standard splits [18] and use Equation (12) as the propagation process. More details of settings are listed in Appendix. Table 9 reports the mean accuracy of each model over 10 runs. Note that we do not include the result of BernNet on ogbn-papers100M as BernNet has a time complexity quadratic to the order $K$ and fails to finish the preprocessing in 24 hours. We can observe that ChebNetII outperforms both datasets,

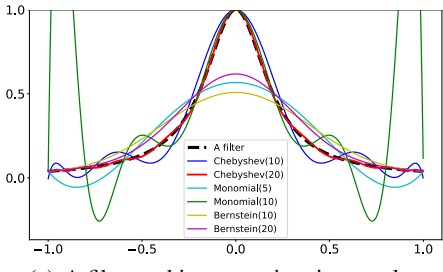

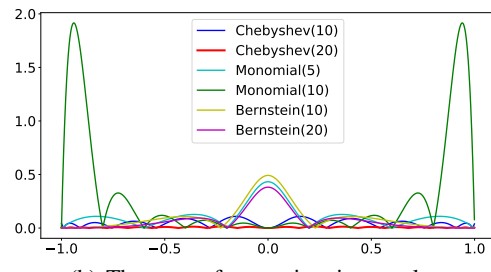

| (a) A filter and its approximation results. | (b) The error of approximation results. |

Figure 2: (a) A Runge filter $h(\hat{\lambda}) = 1/(1 + 25\hat{\lambda}^2)$ and its approximation results by different polynomial bases. (b) The errors of the different approximation results.

which we attribute to Chebyshev Interpolation's superior approximation quality. These results also show that ChebNetII has lesser complexity and greater scalability than BernNet.

### 4.4 Comparison of Different Polynomial Bases

We perform numerical studies comparing the Chebyshev basis to the Monomial and Bernstein bases to demonstrate ChebNetII's approximation power. Considering a Runge filter $h(\hat{\lambda}) = 1/(1+25\hat{\lambda}^2), \hat{\lambda} \in [-1, 1]$, Figures 2(a) and 2(b) depict the approximation results and errors for several polynomial bases, with the polynomial degree $K$ denoted by the numbers in brackets. We find that the Chebyshev basis has a faster convergence rate than the Bernstein basis and does not exhibit the Runge phenomenon compared to the Monomial basis. Notably, JacobiConv [40] investigated different polynomial bases at the same period and discovered that orthogonal polynomial bases could learn graph filters more effectively. These findings provide empirical motivations for designing GNNs with Chebyshev interpolation.

## 5 Conclusion

This paper revisits the problem of approximating spectral graph convolutions with Chebyshev polynomials. We show that ChebNet's inferior performance is primarily due to illegal coefficients learned by approximating analytic filter functions, which leads to over-fitting. Moreover, we propose ChebNetII, a new GNN model based on Chebyshev interpolation, enhancing the original Chebyshev polynomial approximation while reducing the Runge phenomenon. Experiments show that ChebNetII outperforms SOTA methods in terms of effectiveness on real-world both homophilic and heterophilic datasets. For future work, a promising direction is further to improve the performance of ChebNetII on large graphs and investigate the scalability of spectral-based GNNs.

## Acknowledgments and Disclosure of Funding

The work was partially done at Gaoling School of Artificial Intelligence, Peng Cheng Laboratory, Beijing Key Laboratory of Big Data Management and Analysis Methods and MOE Key Lab of Data Engineering and Knowledge Engineering. This research was supported in part by the major key project of PCL (PCL2021A12), by National Natural Science Foundation of China (No. 61972401, No. 61932001, No. 61832017, No. U2001212), by Beijing Natural Science Foundation (No. 4222028), by Beijing Outstanding Young Scientist Program No. BJJWZYJH012019100020098, by Alibaba Group through Alibaba Innovative Research Program, by CCF-Baidu Open Fund (NO.2021PP15002000) and by Huawei-Renmin University joint program on Information Retrieval. We wish to acknowledge the discussion with Professor Zhouchen Lin from Peking University. We also wish to acknowledge the support provided by Engineering Research Center of Next-Generation Intelligent Search and Recommendation, Ministry of Education. Additionally, we acknowledge the support from Intelligent Social Governance Interdisciplinary Platform, Major Innovation & Planning Interdisciplinary Platform for the "Double-First Class" Initiative, Public Policy and Decision-making Research Lab, Public Computing Cloud, Renmin University of China.

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
