# OpenReview forum: "Convolutional Neural Networks on Graphs with Chebyshev Approximation, Revisited"
_NeurIPS.cc/2022/Conference — NeurIPS 2022 Accept_

### Official Review · Reviewer_s9em · 2022-07-08

**Rating:** 6
**Confidence:** 3
**Soundness:** 3 good
**Presentation:** 3 good
**Contribution:** 2 fair

**Summary:**

This paper proposes an extension to ChebNet by introducing Chebyshev Interpolation. The proposed methods is simple, yet provides theoretical support and improved performances for node classification.

**Questions:**

Can ChebNetII be extended with any adaptive learning such as in GPRGNN?

Compared to GPRGNN or MixHop, does ChebNetII learn by convolution using higher-order hops from a node?


Experiments are descent, however, dataset are a bit old. There is a new homophily measure and new non-homophilic benchmark datasets introduced by [Lim et al., 2021a, Lim et al., 2021b].
Can the authors provide few more experiment with the new new non-hohopnilic datasets?

Can the authors include LINK [Zheleva and Getoor, 2009] as a baseline methods?  LINK only uses from the adjacency matrix and gives good indication to learning without node features and convolutions.

Reference:

[Zheleva and Getoor, 2009] Zheleva, E. and Getoor, L. (2009). To join or not to join: The illusion of privacy in  social networks with mixed public and private user profiles. WWW ’09.

[Lim et al., 2021a] Lim, D., Hohne, F. M., Li, X., Huang, S. L., Gupta, V., Bhalerao, O. P., and Lim, S.-N.  (2021a). Large scale learning on non-homophilous graphs: New benchmarks and strong simple methods. In  NeurIPS.

[Lim et al., 2021b] Lim, D., Li, X., Hohne, F., and Lim, S.-N. (2021b). New benchmarks for learning on  non-homophilous graphs. Workshop on Graph Learning Benchmarks, WWW 2021.

**Limitations:**

Limitations and potential negative societal impact are not well expressed.

**Strengths And Weaknesses:**

Strengths:
1.) Simple model with theoretical support
2.) Experimental results are favorable.
3.) Paper is well written and easy to read.

Weaknesses:
1.) Theoretical comparisons to adaptive learning of GPRGNN is not clear.
2.) Incremental work though the contribution is useful.

---

> ### Author Response · Authors · 2022-08-02
> **Response II**
>
> **Q4**: Experiments are descent, however, dataset are a bit old. There is a new homophily measure and new non-homophilic benchmark datasets introduced by [Lim et al., 2021a, Lim et al., 2021b]. Can the authors provide few more experiment with the new new non-hohopnilic datasets?
>
> **A4**: In Tables 1 and 2, we report the results of the new heterophilic graphs. We establish the experimental setting following [Lim et al., 2021a] and use the published results. In Table 1, we find that **ChebNetII outperforms all other methods on 4 out of 5 datasets and the polynomial-based methods on arXiv-year**. Notably, LINK and LINKX outperform ChenNetII on arXiv-year because they use a directed graph on this dataset. Using the directed graphs to the spectral-based GNNs (e.g., BernNet and ChebNetII) is a future meaningful work because the current spectral graph theory only applies to undirected graphs. In Table 2, **ChebNetII obtains a new state-of-the-art result on a large-scale heterophilic graph wiki**. We attribute that ChebNetII can precompute $\hat{\mathbf{L}}^k\mathbf{X}$ (Section 4.3) without the graph sampling used in LINK and LINKX and has a strong approximation ability for filters.
>
> **Table 1**: Experimental results of new non-homophilic benchmark datasets.
>
> | |  Penn94 |    pokec   |   arXiv-year  |    genius   |    twitch-gamers  |
> |:-------|:----------|:----------|:-----:|:----------:|:----------:|
> | LINK       | 80.79±0.49 | 80.54±0.03 | 53.97±0.18 | 73.56±0.14 | 64.85±0.21 |
> | LINKX      | 84.71±0.52 | 82.04±0.07 | **56.00±1.34** | 90.77±0.27 | 66.06±0.19 |
> | GCN        | 82.47±0.27 | 75.45±0.17 | 46.02±0.26  | 87.42±0.37 | 62.18±0.26 |
> | ChebNet    | 82.59±0.31 | 72.71±0.66 | 46.76±0.24 | 89.36±0.31 | 62.31±0.37 |
> | GPR-GNN | 81.38±0.16 | 78.83±0.05 |  45.07±0.21 | 90.05±0.31 | 61.89±0.29 |
> | BernNet   | 83.79±0.39 | 81.67±0.07 |  46.34±0.32 | 90.47±0.33 | 65.16±0.22 |
> | ChebNetII | **84.91±0.45** | **82.33±0.28** |  48.53±0.31 | **90.85±0.32** | **66.12±0.25** |
>
> **Table 2**: Experimental results of large non-homophilic graph wiki (1.9M nodes and 303.4M edges).
>
> |            |  MLP |    LINK  |   LINKX   |    GPR-GNN   |    ChebNetII   |
> |:--------  |:----------:|:----------:|:--------------:|:----------:|:----------:|
> | wiki       | 37.38±0.21 | 57.11±0.26 | 59.80±0.41 | 58.73±0.34 | **60.95±0.39** |
>
> **Q5**: Can the authors include LINK [Zheleva and Getoor, 2009] as a baseline methods? LINK only uses from the adjacency matrix and gives good indication to learning without node features and convolutions.
>
> **A5**: In Tables 3 and 4, we present the mean node classification accuracy of semi-/full-supervised node classification for LINK and ChebNetII. **ChebNetII outperforms LINK on most datasets, especially for the semi-supervised task**. LINK attains good results on Chameleon and Squirrel for the full-supervised task, possibly because the graph structures of these two datasets are more crucial for the downstream task. Further investigation beyond the scope of this work.
>
> **Table 3**: Mean classification accuracy of **semi-supervised** node classification with random splits.
>
> |              |  Chamaleon |    Squirrel   |   Texas     |    Cornell   |    Actor   |     Cora   |   Citeseer|Pubmed|
> |:-------------|:----------:|:----------:|:--------------:|:----------:|:----------:|:----------:|:----------:|:----------:|
> | LINK       | 42.81±1.32 | 30.37±1.68 | 39.71±9.13 | 33.46±10.17 | 20.48±0.44 |49.58±3.98 |27.37±3.01 |44.94±0.95|
> | ChebNetII       | **43.42±3.54** | **33.96±1.22** | **46.58±7.68** | **42.19±11.61** | **30.18±0.81** |**82.42±0.64** |**69.89±1.21**|**79.51±1.03**|
>
> **Table 4**: Mean classification accuracy of **full-supervised** node classification with random splits.
>
> |             |  Chamaleon |    Squirrel   |   Texas     |    Cornell   |    Actor   |     Cora   |   Citeseer|Pubmed|
> |:-------------|:----------:|:----------:|:--------------:|:----------:|:----------:|:----------:|:----------:|:----------:|
> | LINK       | 71.09±1.24 | **65.13±0.71** | 70.81±4.42 | 54.09±4.91 | 23.97±0.55 |80.81±0.96 |61.10±1.28 |81.63±0.33|
> | ChebNetII       | **71.37±1.01** | 57.72±0.59 | **93.28±1.47** | **92.30±1.48** | **41.75±1.07** |**88.71±0.93** |**80.53±0.79**|**88.93±0.29**|
>
> We will gladly answer any additional questions you may have.

---

> > ### Comment · Reviewer_s9em · 2022-08-09
> > **Reply**
> >
> > Thank you for the update on my questions!.
> >
> > Please include the experimental results for the new experiments for [Lim et al., 2021a, Lim et al., 2021b] datasets in the paper. I have updated my score.

---

> > > ### Author Response · Authors · 2022-08-09
> > > **Re**
> > >
> > > Thank you for your helpful feedback and for supporting our work! We will include the new results in the final version of the paper.

---

> ### Author Response · Authors · 2022-08-02
> **Response I**
>
> Thank you for your insightful feedback.
>
> **Q1**: Can ChebNetII be extended with any adaptive learning such as in GPRGNN? Theoretical comparisons to adaptive learning of GPRGNN is not clear.
>
> **A1**: We want to clarify that ChebNetII has an adaptive learning ability for different datasets similar to GPR-GNN, which is stated as "ChebNetII can learn arbitrary graph filters" in the paper. In fact, ChebNetII has better learning and approximation capabilities than GPR-GNN. Specifically, ChebNetII has the following advantages over GPRGNN.
> 1.  ChebNetII provides a Near-minimax approximation for a filter function (i.e., Theorem 3.1).
> 2.  ChebNetII can reduce the Runge phenomenon, a significant obstacle for approximating filter functions by the monomial basis (i.e., Theorem 3.3).
> 3.  ChebNetII can more easily impose constraints on the learned filters, such as obtaining non-negative filters, a requirement proposed in BernNet.
>
> We will include more details about the comparisons of ChebNetII and GPR-GNN in the final version of the paper to assist readers in comprehending.
>
> **Q2**: Compared to GPRGNN or MixHop, does ChebNetII learn by convolution using higher-order hops from a node?
>
> **A2**: Yes, ChebNetII can use higher-orde hops to learn the graph convolution. In Equation (8) (i.e., $\mathbf{Y}=\frac{2}{K+1} \sum\limits_{k=0}^{K} \sum\limits_{j=0}^{K} \gamma_j T_k(x_j)T_k(\hat{\mathbf{L}})f_{\theta}(\mathbf{X})$), $T_k(\hat{\mathbf{L}})$ denotes the Chebyshev polynomial for the scaled Laplacian matrix $\hat{\mathbf{L}}$. As $k$ increases, ChebNetII uses a higher-order scaled Laplacian matrix (i.e., $(\hat{\mathbf{L}})^k$) for propagation, that is using higher-order hops to learn the graph convolution.
>
>
> **Q3**: Incremental work though the contribution is useful.
>
> **A3**:  Although our work appears to be an incremental improvement on ChebNet, it is more of **an exploration of how the polynomial approximation theory might be used for GNNs**. There are two primary motivations:
> 1. ChebNet was published before GCN and is more expressive than GCN (GCN is a simplification of ChebNet), but ChebNet's performance is inferior to GCN's. So **it is essential to understand why ChebNet does not perform well and how to improve it**.
> 2. GPR-GNN and BernNet empirically demonstrate that the Monomial and Bernstein bases outperform the Chebyshev basis in learning the spectral graph convolutions. Such conclusions are counter-intuitive in the field of approximation theory, where it is established that the Chebyshev polynomial achieves the optimum convergent rate for approximating a function. So **it is crucial to know why is ChebNet has a higher approximation ability, but its performance is inferior to that of GPR-GNN and BernNet**.
>
> These problems are solved in our work. We summarize the work's contributions as follows:
> 1. We first **revisit the fundamental problem of approximating the spectral graph filters with Chebyshev polynomials** and observe that the coefficients of the Chebyshev expansion for an analytic filter function need to satisfy an inevitable convergence (Theorem 2.1).
> 2. We **demonstrate that ChebNet's inferior performance** is primarily due to illegal coefficients learned by ChebNet approximating analytic filter functions, which leads to over-fitting (Section 2.3).
> 3. We **propose ChebNetII, a novel GNN model** based on Chebyshev interpolation, which enhances the original Chebyshev polynomial approximation while reducing the Runge phenomenon (Section 3.2).
> 4. We **theoretically evaluate ChebNetII's approximation performance** with related models (e.g., GPR-GNN and BernNet), showing that ChebNetII has a strong capability for approximating a filter function (Section 3.3).
> 5. We conduct an extensive experimental study to demonstrate that ChebNetII can achieve superior performance on real-world datasets, **particularly for semi-supervised tasks and scalability** (Section 4).

---

### Official Review · Reviewer_jWTu · 2022-07-10

**Rating:** 6
**Confidence:** 4
**Soundness:** 3 good
**Presentation:** 3 good
**Contribution:** 3 good

**Summary:**

This paper revisited the Chebyshev approximation and showed that the inferior performance of ChebNet is due to illegal learnt coefficients. On the basis of this insight, a new GNN model called ChebNetII is proposed enhancing the Chebyshev polynomial approximation and reducing the Runge phenomenon. ChebNetII can achieve the state-of- art performance on some open graph datasets.

**Questions:**

1.	Do you have some suggestions about the choice of $K$? Does it the same as the original ChebNet?
2.	What does ‘weakly singular’ mean? The vanishing of the derivative of $f$ seems inequivalent to ‘$f$ be locally given by a convergent power series in the interval $(-1,1)$’ at line 132. Could you please give some clear mathematical explanation?
3.	Why one can assert $w_0=w_1$ implies a high-pass filter and $w_0=-w_1$ at line 102 implies a low-pass filter?

**Limitations:**

CheNetII seems to have broad applications besides the experiments shown in the paper. One can show more future directions in the paper.

**Strengths And Weaknesses:**

*Strengths: The authors showed a sharp eye for the subtle inferior performance of ChebNet compared with GCN, GPR-GNN and BernNet and proposed two questions to generalize the problem. By citing related theorems, both the questions are explained clearly. The novel model ChebNet resolves the questions without introducing expensive additional computation complexity.
*Weaknesses: The main theoretical frame is assembled by existing results. It is okay but maybe it makes the paper not very innovative.

---

> ### Author Response · Authors · 2022-08-02
> **Response**
>
> We appreciate your insightful feedback.
>
> **Q1**: The main theoretical frame is assembled by existing results. It is okay but maybe it makes the paper not very innovative.
>
> **A1**: Yes, we use several known results from the polynomial approximation theory. However, our goal is to **explain and improve the spectral-based GNNs using the existing mathematical theory**, not to create a new polynomial approximation theory. Below are the two key things we do:
> 1. We use the polynomial approximation theory to explain why ChebNet is poor.
> 2. We use the Chebyshev interpolation theory to obtain a good ChebNet.
>
> Neither of these things has been done in the GNN field, so we argue that **our work is actually a novel exploration of how the polynomial approximation theory improves existing GNN models**.
>
> **Q2**:  Do you have some suggestions about the choice of $K$? Does it the same as the original ChebNet?
>
> **A2**: The Chebyshev interpolation theory states that the error of ChebNetII decreases as $K$ increases, but in practice, it depends on how well ChebNetII learns $\gamma_j$'s from the training data. For convenience and fairness of comparison, we use the same settings as APPNP, GPR-GNN, and BernNet (i.e., $K=10$). Notably,  each $k$ ($k \in {0,1,2,\cdots,K})$  of ChebNet corresponds to a weight matrix $\mathbf{W}_k$, with larger $K$ prone to overfitting. ChebNetII, which utilizes coefficients $\gamma_j$ rather than $\mathbf{W}_k$, eliminates this issue. Further investigation regarding the choice of $K$ is meaningful, but it is beyond the scope of this work.
>
> **Q3**: What does ‘weakly singular’ mean? The vanishing of the derivative of $f$ seems inequivalent to ‘$f$ be locally given by a convergent power series in the interval $(-1,1)$’ at line 132. Could you please give some clear mathematical explanation?
>
> **A3**: The "weakly singular" means that the derivative of $f$ vanishes **at the boundaries**  but **analytic** everywhere in the interval (-1,1). For example, we consider a function $u(x)=(1-x^2)\ln(1-x^2),x\in [-1,1]$, which is weakly singular and not derivable at endpoints $\pm1$. According to [46], we can derive $q = 3$ in Theorem 2.1 that is Chebyshev coefficients $w_k$ of $u(x)$ will asymptotically (as $k \to \infty$) decrease proportionally to $k^{-3}$. Notably, $f$ can be locally given by a convergent power series in the interval $(-1,1)$ due to $f$ is **analytic** in the interval $(-1,1)$. For more details of Theorem 2.1 in the paper, you can see Lemma 1 and Theorem 1 in [46]. We will also add more details in the final version of the paper for the reader to understand.
>
> **Q4**: Why one can assert $w_0=w_1$ implies a high-pass filter and $w_0=-w_1$ at line 102 implies a low-pass filter?
>
> **A4**: Lines 101 and 102 in the paper are 'If we set $K = 1$ and $w_0=w_1$ in the Equation (2) (i.e., $\mathbf{y}\approx\sum\limits_{k=0}^{K}w_kT_k(\hat{\mathbf{L}})\mathbf{x}$), ChebNet corresponds to a high-pass filter; if we set $K = 1$ and $w_0=-w_1$, ChebNet becomes a low-pass filter.'
>
> 1. When we set $K = 1$ and $w_0=w_1$ in Equation (2), $\mathbf{y}\approx w\mathbf{I}x+w\hat{\mathbf{L}}x=w\frac{2\mathbf{L}}{\lambda_{max}}x$, where $w=w_0=w_1$ and $\hat{\mathbf{L}}=2\mathbf{L}/\lambda_{max}-\mathbf{I}$. The corresponding filter is $h(\lambda)=\frac{2\lambda}{\lambda_{max}}$, which is a linear high-pass filter.
>
> 2. When we set $K = 1$ and $w_0=-w_1$ in Equation (2), $\mathbf{y}\approx w\mathbf{I}x-w\hat{\mathbf{L}}x=w(2\mathbf{I}-\frac{2\mathbf{L}}{\lambda_{max}})x$, where $w=w_0=-w_1$ and $\hat{\mathbf{L}}=2\mathbf{L}/\lambda_{max}-\mathbf{I}$. The corresponding filter is $h(\lambda)=2-\frac{2\lambda}{\lambda_{max}}$, which is a linear low-pass filter.
>
> We will gladly answer any additional questions you may have.

---

### Official Review · Reviewer_ypP8 · 2022-07-10

**Rating:** 8
**Confidence:** 5
**Soundness:** 4 excellent
**Presentation:** 4 excellent
**Contribution:** 4 excellent

**Summary:**

This paper revisits the traditional problem of approximating graph convolutional networks using Chebyshev polynomials. The paper shows that the performance of the network can be greatly improved by using Chebyshev Interpolation to replace the coefficients of ChebNet on both homophilous and heterophilous graphs.

**Questions:**

The questions have been discussed in detail as above. To sum up, the main questions are:

•	The difference between ChebBase and ChebNet should be better explained.

•	It would be interesting to showcase some of the filters that ChebNetII has learned. What distinguishes them from the ones learned by BernNet?

•	The best results should also be highlighted in Table 3.


**Limitations:**

Yes,  the limitation of this work has been well discussed.

**Strengths And Weaknesses:**

This work is mostly well-written with informative content and new ideas that are worth publishing.

Strengths:

•	The use of Chebyshev Interpolation is both theoretically and practically justified.

•	ChebNetII's performance improvement is significant, particularly for semi-supervised node classification tasks.

•	Following on from the second point, to the best of my knowledge, this paper presents the first empirical study of semi-supervised node classification with polynomial graph filters.

•	ChebNetII scales to the graph Papers100M, which has more than a billion edges.

•	The overall presentation is satisfactory.

Weaknesses:

•	There should be a clearer explanation of the distinction between ChebBase and ChebNet.

•	It would be interesting to showcase some of the filters that ChebNetII has learned. What distinguishes them from the ones learned by BernNet?

Minor points:

•	The best results should also be highlighted in Table 3.

---

> ### Author Response · Authors · 2022-08-02
> **Response**
>
> We appreciate your insightful feedback.
>
> **Q1**: The difference between ChebBase and ChebNet  should be better explained.
>
> **A1**: The propagation processes of ChebNet and ChebBase are described in Equations (3) and (5) in the paper. That is
>
> 1) $\mathbf{Y}=\sum\nolimits_{k=0}^{K}T_k(\hat{\mathbf{L}})\mathbf{X}\mathbf{W}_k $ for ChebNet,
>
> 2) $\mathbf{Y} = \sum\nolimits_{k=0}^{K}w_k T_k(\hat{\mathbf{L}}) f_{\theta}\left(\mathbf{X}\right)$ for ChebBase.
>
> The Chebyshev coefficients of ChebNet are implicitly encoded in the weight matrices $\mathbf{W}_k$, making it challenging to add constraints on the coefficients. On the other hand, ChebBase uses a propagation process similar to that of GPR-GNN and BernNet, which explicitly learn the Chebyshev coefficients and make it simple to add constraints to them. Notably, ChebNet and ChebBase employ the same filtering operation (i.e., Equation (2) in the paper). We will revise the text and include more details about the difference between ChebBase and ChebNet to assist readers in comprehending it.
>
> **Q2**:  It would be interesting to showcase some of the filters that ChebNetII has learned. What distinguishes them from the ones learned by BernNet?
>
> **A2**:  In Tables 1 and 2, we report the filter responses ( $\lambda$ denotes the eigenvalue of the normalized Laplacian matrix $\mathbf{L}$) learned by ChebNetII and BernNet on Chameleon for the semi- /full-supervised node classification task. For the semi-supervised task, we observe that BernNet learns an ineffective filter, which performs poorly and is similar to the initialization (an all-pass filter). On the other hand, ChebNetII learns a band-rejection pass filter and performs better. We attribute the effective filter learned by ChebNetII to **the faster convergence rate of the ChebNetII for approximating a filter function** (i.e., Theorem 3.2 in the paper). For the full-supervised task, ChebNetII and BernNet both learn a comb-alike filter. Notably, the shape of the filter learned by ChebNetII is relatively more complex, demonstrating that ChebNetII has a stronger approximating ability for learning filters. We appreciate your advice and report more comparisons in the revised supplementary material.
>
> **Table 1**: The filter responses learned by BernNet and ChebNetII on Chameleon for the **semi-supervised** learning.
>
> | $\lambda$ |  0.0 |  0.2 |  0.4 |  0.6 |  0.8 |  1.0 |  1.2 |  1.4 |  1.6 |  1.8 |  2.0 |
> |-----------|:----:|:----:|:----:|:----:|:----:|:----:|:----:|:----:|:----:|:----:|:----:|
> | BernNet   | 1.0098 | 1.0211 | 1.0207 | 1.0193 | 1.0157 | 1.0133 | 1.0108 | 1.0086 | 1.0056 | 1.0017 | 1.0001 |
> | ChebNetII  | 1.0103 | 0.7324 | 0.4489 | 0.2914 | 0.1019 | 0.0007 | 0.1131 | 0.2906 | 0.4585 | 0.7173 | 0.9879 |
>
> **Table 2**: The filter responses learned by BernNet and ChebNetII on Chameleon for the **full-supervised** learning.
>
> | $\lambda$ |  0.0 |  0.2 |  0.4 |  0.6 |  0.8 |  1.0 |  1.2 |  1.4 |  1.6 |  1.8 |  2.0 |
> |-----------|:----:|:----:|:----:|:----:|:----:|:----:|:----:|:----:|:----:|:----:|:----:|
> | BernNet   | 1.7429 | 1.9151 | 1.5892 | 1.0913 | 0.7871 | 0.5935 | 0.8239 | 1.1762 | 1.556 | 1.6491 | 1.5176 |
> | ChebNetII | 1.4263 | 1.5982 | 1.1734 | 1.2758 | 0.7231 | 0.3762 | 0.9231 | 1.2065 | 1.1137 | 1.5845 | 1.7692 |
>
> **Q3**: The best results should also be highlighted in Table 3.
>
> **A3**: We will highlight the best results in Table 3 in the final version of the paper. We appreciate the advice!
>
> We will gladly answer any additional questions you may have.

---

> > ### Comment · Reviewer_ypP8 · 2022-08-09
> > **No further questions**
> >
> > The authors have addressed all the issues raised in my previous comments. I am happy to hold the evaluation.

---

> > > ### Author Response · Authors · 2022-08-09
> > > **Re**
> > >
> > > Thank you for your useful feedback and for supporting our work!

---

### Author Response · Authors · 2022-08-08
**General comment to reviewers**

We want to thank each reviewer for their helpful comments. We have answered each question in the comment sections and will gladly answer any additional questions you may have.

---

### Meta-Review · Area_Chair_3aPc · 2022-08-23

**Recommendation:** Accept
**Confidence:** Certain

**Metareview:**

The authors consider the traditional problem of approximating graph convolutional networks using Chebyshev polynomial, which is known as Chebnet. Then, the authors propose a new GNN model called ChebNetII enhancing for reducing the Runge phenomenon; this is an important contribution to GNN. Overall, the reviewers are positive about the paper. Thus, I also vote for acceptance.

**Award:**

No

---

### Decision · Program_Chairs · 2022-09-14

Accept